# Deep Generative Models for Distribution-Preserving Lossy Compression

**Michael Tschannen**
ETH Zürich
michaelt@nari.ee.ethz.ch

**Eirikur Agustsson**
Google AI Perception
eirikur@google.com

**Mario Lucic**
Google Brain
lucic@google.com

## Abstract

We propose and study the problem of distribution-preserving lossy compression. Motivated by recent advances in extreme image compression which allow to maintain artifact-free reconstructions even at very low bitrates, we propose to optimize the rate-distortion tradeoff under the constraint that the reconstructed samples follow the distribution of the training data. The resulting compression system recovers both ends of the spectrum: On one hand, at zero bitrate it learns a generative model of the data, and at high enough bitrates it achieves perfect reconstruction. Furthermore, for intermediate bitrates it smoothly interpolates between learning a generative model of the training data and perfectly reconstructing the training samples. We study several methods to approximately solve the proposed optimization problem, including a novel combination of Wasserstein GAN and Wasserstein Autoencoder, and present an extensive theoretical and empirical characterization of the proposed compression systems.

## 1   Introduction

Data compression methods based on deep neural networks (DNNs) have recently received a great deal of attention. These methods were shown to outperform traditional compression codecs in image compression  [1–10], speech compression [11], and video compression [12] under several distortion measures. In addition, DNN-based compression methods are flexible and can be adapted to specific domains leading to further reductions in bitrate, and promise fast processing thanks to their internal representations that are amenable to modern data processing pipelines [13].

In the context of image compression, learning-based methods arguably excel at low bitrates by learning to realistically synthesize local image content, such as texture. While learning-based methods can lead to larger distortions w.r.t. measures optimized by traditional compression algorithms, such as peak signal-to-noise ratio (PSNR), they avoid artifacts such as blur and blocking, producing visually more pleasing results [1–10]. In particular, visual quality can be improved by incorporating generative adversarial networks (GANs) [14] into the learning process [4, 15]. Work [4] leveraged GANs for artifact suppression, whereas [15] used them to learn synthesizing image content beyond local texture, such as facades of buildings, obtaining visually pleasing results at very low bitrates.

In this paper, we propose a formalization of this line of work: *A compression system that respects the distribution of the original data at all rates*—a system whose decoder generates i.i.d. samples from the data distribution at zero bitrate, then gradually produces reconstructions containing more content of the original image as the bitrate increases, and eventually achieves perfect reconstruction at high enough bitrate (see Figure 1 for examples). Such a system can be learned from data in a fully unsupervised fashion by solving what we call the *distribution-preserving lossy compression (DPLC)* problem: Optimizing the rate-distortion tradeoff under the constraint that the reconstruction follows the distribution of the training data. Enforcing this constraint promotes artifact-free reconstructions, at all rates.

| 0.000 | 0.008 | 0.031 | 0.125 | 0.500 | original | | 0.000 | 0.008 | 0.031 | 0.125 | 0.500 | original |

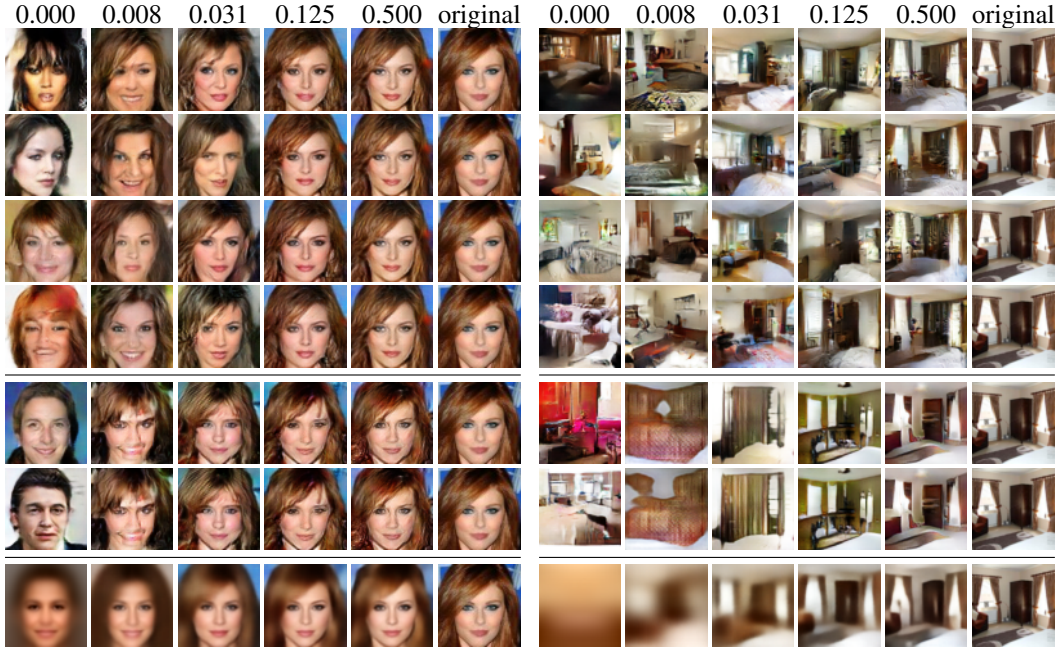

Figure 1: Example (testing) reconstructions for CelebA (left) and LSUN bedrooms (right) obtained by our DPLC method based on Wasserstein++ (rows 1–4), global generative compression GC [15] (rows 5–6), and a compressive autoencoder (CAE) baseline (row 7), as a function of the bitrate (in bits per pixel). We stress that as the bitrate decreases, DPLC manages to generate diverse and realistic-looking images, whereas GC struggles to produce diverse reconstructions and the CAE reconstructions become increasingly blurry.

We then show that the algorithm proposed in [15] is solving a special case of the DPLC problem, and demonstrate that it fails to produce stochastic decoders as the rate tends to zero in practice, i.e., it is not effective in enforcing the distribution constraint at very low bitrates. This is not surprising as it was designed with a different goal in mind. We then propose and study different alternative approaches based on deep generative models that overcome the issues inherent with [15]. In a nutshell, one first learns a generative model and then applies it to learn a stochastic decoder, obeying the distribution constraint on the reconstruction, along with a corresponding encoder. To quantify the distribution mismatch of the reconstructed samples and the training data in the learning process we rely on the Wasserstein distance. One distinct advantage of our approach is that we can theoretically characterize the distribution of the reconstruction and bound the distortion as a function of the bitrate.

On the practical side, to learn the generative model, we rely on Wasserstein GAN (WGAN) [16] and Wasserstein autoencoder (WAE) [17], as well as a novel combination thereof termed Wasserstein++. The latter attains high sample quality comparable to WGAN (when measured in terms of the Fréchet inception distance (FID) [18]) and yields a generator with good mode coverage as well as a structured latent space suited to be combined with an encoder, like WAE. We present an extensive empirical evaluation of the proposed approach on two standard GAN data sets, CelebA [19] and LSUN bedrooms [20], realizing the first system that effectively solves the DPLC problem.

**Outline.** We formally define and motivate the DPLC problem in Section 2. We then present several approaches to solve the DPLC problem in Section 3. Practical aspects are discussed in Section 4 and an extensive evaluation is presented in Section 5. Finally, we discuss related work in Section 6.

## 2  Problem formulation

**Notation.** We use uppercase letters to denote random variables, lowercase letters to designate their values, and calligraphy letters to denote sets. We use the notation $P_X$ for the distribution of the random variable $X$ and $\mathbb{E}_X[X]$ for its expectation. The relation $W \sim P_X$ designates that $W$ follows the distribution $P_X$, and $X \sim Y$ indicates that $X$ and $Y$ are identically distributed.

**Setup.** Consider a random variable $X \in \mathcal{X}$ with distribution $P_X$. The latter could be modeling, for example, natural images, text documents, or audio signals. In standard lossy compression, the goal is to create a rate-constrained *encoder* $E \colon \mathcal{X} \to \mathcal{W} := \{1, \ldots, 2^R\}$, mapping the *input* to a *code* of $R$ bits, and a *decoder* $D \colon \mathcal{W} \to \mathcal{X}$, mapping the code back to the input space, such as to minimize some distortion measure $d \colon \mathcal{X} \times \mathcal{X} \to \mathbb{R}_+$. Formally, one aims at solving

$$\min_{E,D} \quad \mathbb{E}_X[d(X, D(E(X)))]. \tag{1}$$

In the classic lossy compression setting, both $E$ and $D$ are typically deterministic. As a result, the number of distinct reconstructed inputs $\hat{X} := D(E(X))$ is bounded by $2^R$. The main drawback is, as $R$ decreases, the reconstruction $\hat{X}$ will incur increasing degradations (such as blur or blocking in the case of natural images), and will be constant for $R = 0$. Note that simply allowing $E, D$ in (1) to be stochastic does not resolve this problem as discussed in Section 3.

**Distribution-preserving lossy compression.** Motivated by recent advances in extreme image compression [15], we propose and study a novel compression problem: Solve (1) under the constraint that the distribution of reconstructed instances $\hat{X}$ follows the distribution of the training data $X$. Formally, we want to solve the problem

$$\min_{E,D} \quad \mathbb{E}_{X,D}[d(X, D(E(X)))] \quad \text{s.t.} \quad D(E(X)) \sim X, \tag{2}$$

where the decoder is allowed to be stochastic.[1] The goal of the distribution matching constraint is to enforce artifact-free reconstructions at all rates. Furthermore, as the rate $R \to 0$, the solution converges to a generative model of $X$, while for sufficiently large rates $R$ the solution guarantees perfect reconstruction and trivially satisfies the distribution constraint.

## 3 Deep generative models for distribution-preserving lossy compression

The distribution constraint makes solving the problem (2) extremely challenging, as it amounts to learning an exact generative model of the generally unknown distribution $P_X$ for $R = 0$. As a remedy, one can relax the problem and consider the regularized formulation,

$$\min_{E,D} \quad \mathbb{E}_{X,D}[d(X, D(E(X)))] + \lambda d_f(P_{\hat{X}}, P_X), \tag{3}$$

where $\hat{X} = D(E(X))$, and $d_f$ is a (statistical) divergence that can be estimated from samples using, e.g., the GAN framework [14].

**Challenges of the extreme compression regime.** At any finite rate $R$, the distortion term and the divergence term in (3) have strikingly opposing effects. In particular, for distortion measures for which $\min_y d(x, y)$ has a unique minimizer for every $x$, the decoder minimizing the distortion term is constant, conditioned on the code $w$. For example, if $d(x, y) = \|x - y\|^2$, the optimal decoder $D$ for a fixed encoder $E$ obeys $D(w) = \mathbb{E}_X[X|E(X) = w]$, i.e., it is biased to output the mean. For many popular distortions measures, $D, E$ minimizing the distortion term therefore produce reconstructions $\hat{X}$ that follow a *discrete* distribution, which is at odds with the often *continuous* nature of the data distribution. In contrast, the distribution divergence term encourages $D \circ E$ to generate outputs that are as close as possible to the data distribution $P_X$, i.e., it encourages $D \circ E$ to follow a continuous distribution if $P_X$ is continuous. While in practice the distortion term can have a stabilizing effect on the optimization of the divergence term (see [15]), it discourages the decoder form being stochastic—the decoder learns to ignore the noise fed as an input to provide stochasticity, and does so even when adjusting $\lambda$ to compensate for the increase in distortion when $R$ decreases (see the experiments in Section 5). This is in line with recent results for deep generative models in conditional settings: As soon as they are provided with context information, they tend to ignore stochasticity as discussed in [21, 22], and in particular [23] and references therein.

**Proposed method.** We propose and study different generative model-based approaches to approximately solve the DPLC problem. These approaches overcome the aforementioned problems and can be applied for all bitrates $R$, enabling a gentle tradeoff between matching the distribution of the training data and perfectly reconstructing the training samples. Figure 2 provides an overview of the proposed method.

In order to mitigate the bias-to-the-mean-issues with relaxations of the form (3), we decompose $D$ as $D = G \circ B$, where $G$ is a generative model taking samples from a fixed prior distribution $P_Z$ as an input, trained to minimize a divergence between $P_{G(Z)}$ and $P_X$, and $B$ is a stochastic function that is trained together with $E$ to minimize distortion for a fixed $G$.

Out of the plethora of divergences commonly used for learning generative models $G$ [14, 24], the Wasserstein distance between $P_{\hat{X}}$ and $P_X$ is particularly well suited for DPLC. In fact, it has a distinct advantage as it can be defined for an arbitrary transportation cost function, in particular for the distortion measure $d$ quantifying the quality of the reconstruction in (2). For this choice of transportation cost, we can *analytically* quantify the distortion as a function of the rate and the Wasserstein distance between $P_{G(Z)}$ and $P_X$.

**Learning the generative model** $G$**.** The Wasserstein distance between two distributions $P_X$ and $P_Y$ w.r.t. the measurable cost function $c \colon \mathcal{X} \times \mathcal{X} \to \mathbb{R}_+$ is defined as

$$W_c(P_X, P_Y) := \inf_{\Pi \in \mathcal{P}(P_X, P_Y)} \mathbb{E}_{(X,Y) \sim \Pi}[c(X,Y)], \tag{4}$$

where $\mathcal{P}(P_X, P_Y)$ is a set of all joint distributions of $(X, Y)$ with marginals $P_X$ and $P_Y$, respectively. When $(\mathcal{X}, d')$ is a metric space and we set $c(x,y) = d'(x,y)$ we have by Kantorovich-Rubinstein duality [25] that

$$W_{d'}(P_X, P_Y) := \sup_{f \in \mathcal{F}_1} \mathbb{E}_X[f(X)] - \mathbb{E}_Y[f(Y)], \tag{5}$$

where $\mathcal{F}_1$ is the class of bounded 1-Lipschitz functions $f \colon \mathcal{X} \to \mathbb{R}$. Let $G \colon \mathcal{Z} \to \mathcal{X}$ and set $Y = G(Z)$ in (5), where $Z$ is distributed according to the prior distribution $P_Z$. Minimizing the latter over the parameters of the mapping $G$, one recovers the Wasserstein GAN (WGAN) proposed in [16]. On the other hand, for $Y = G(Z)$ with deterministic $G$, (4) is equivalent to factorizing the couplings $\mathcal{P}(P_X, P_{G(Z)})$ through $Z$ using a conditional distribution function $Q(Z|X)$ (with $Z$-marginal $Q_Z(Z)$) and minimizing over $Q(Z|X)$ [17], i.e.,

$$\inf_{\Pi \in \mathcal{P}(P_X, P_{G(Z)})} \mathbb{E}_{(X,Y) \sim \Pi}[c(X,Y)] = \inf_{Q \colon Q_Z = P_Z} \mathbb{E}_X \mathbb{E}_{Q(Z|X)}[c(X, G(Z))]. \tag{6}$$

In this model, the so-called *Wasserstein Autoencoder (WAE)*, $Q(Z|X)$ is parametrized as the push-forward of $P_X$, through some possibly stochastic function $F \colon \mathcal{X} \to \mathcal{Z}$ and (6) becomes

$$\inf_{F \colon F(X) \sim P_Z} \mathbb{E}_X \mathbb{E}_F[c(X, G(F(X)))], \tag{7}$$

which is then minimized over $G$.

Note that, in order to solve (2), one cannot simply set $c(x,y) = d(x,y)$ and replace $F$ in (7) with a rate-constrained version $\hat{F} = B \circ E$, where $E$ is a rate-constrained encoder as introduced in Section 2 and $B \colon \mathcal{W} \to \mathcal{Z}$ a stochastic function. Indeed, the tuple $(X, G(F(X)))$ in (7) parametrizes the couplings $\mathcal{P}(P_X, P_{G(Z)})$ and $G \circ F$ should therefore be of high model capacity. Using $\hat{F}$ instead of $F$ severely constrains the model capacity of $G \circ \hat{F}$ (for small $R$) compared to $G \circ F$, and minimizing (7) over $G \circ \hat{F}$ would hence not compute a $G(Z)$ which approximately minimizes $W_c(P_X, P_{G(Z)})$.

**Learning the function** $B \circ E$**.** To circumvent this issue, instead of replacing $F$ in (7) by $\hat{F}$, we propose to first learn $G^\star$ by either minimizing the primal form (6) via WAE or the dual form (5) via WGAN (if $d$ is a metric) for $c(x,y) = d(x,y)$, and *subsequently* minimize the distortion as

$$\min_{B, E \colon B(E(X)) \sim P_Z} \mathbb{E}_{X,B}[d(X, G^\star(B(E(X))))] \tag{8}$$

w.r.t. the fixed generator $G^\star$. We then recover the stochastic decoder $D$ in (2) as $D = G^\star \circ B$. Clearly, the distribution constraint in (8) ensures that $G^\star(B(E(X))) \sim G^\star(Z)$ since $G$ was trained to map $P_Z$ to $P_X$.

**Reconstructing the Wasserstein distance.** The proposed method has the following guarantees.

**Theorem 1.** *Suppose $\mathcal{Z} = \mathbb{R}^m$ and $\|\cdot\|$ is a norm on $\mathbb{R}^m$. Further, assume that $\mathbb{E}[\|Z\|^{1+\delta}] < \infty$ for some $\delta > 0$, let $d$ be a metric and let $G^\star$ be $K$-Lipschitz, i.e., $d(G^\star(x), G^\star(y)) \leq K\|x - y\|$. Then,*

$$W_d(P_X, P_{G^\star(Z)}) \leq \min_{\substack{B,E \colon \\ B(E(X)) \sim P_Z}} \mathbb{E}_{X,B}[d(X, G^\star(B(E(X))))] \leq W_d(P_X, P_{G^\star(Z)}) + 2^{-\frac{R}{m}} KC, \tag{9}$$

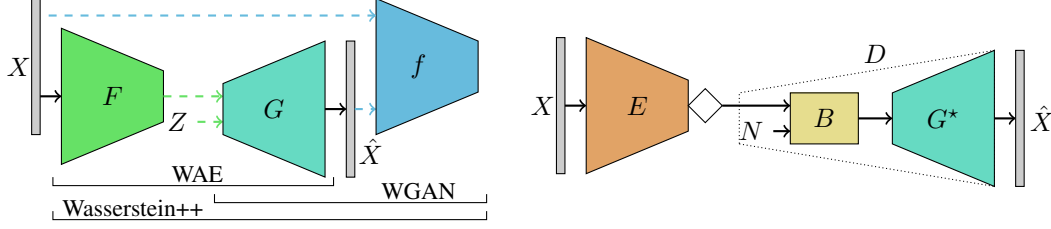

Figure 2: **Left:** A generative model $G$ of the data distribution is commonly learned by minimizing the Wasserstein distance between $P_X$ and $P_{G(Z)}$ either (i) via Wasserstein Autoencoder (WAE) [17], where $G \circ F$ parametrizes the couplings between $P_X$ and $P_{G(Z)}$, or (ii) via Wasserstein GAN (WGAN) [16], which relies on the critic $f$. We propose *Wasserstein++*, a novel approach subsuming both WAE and WGAN. **Right:** Combining the trained generative model $G^\star$ with a rate-constrained encoder $E$ (quantization denoted by $\diamond$-symbol), and a stochastic function $B$ (stochasticity is provided through the noise vector $N$) to realize a distribution-preserving compression (DPLC) system which minimizes the distortion between $X$ and $\hat{X}$, while ensuring that $P_X$ and $P_{\hat{X}}$ are similar at all rates.

*where $C > 0$ is an absolute constant that depends on $\delta, m, \mathbb{E}[\|Z\|^{1+\delta}]$, and $\|\cdot\|$. Furthermore, for an arbitrary distortion measure $d$ and arbitrary $G^\star$ it holds for all $R \geq 0$*

$$W_d(P_X, P_{G^\star(B(E(X)))}) = W_d(P_X, P_{G^\star(Z)}). \tag{10}$$

The proof is presented in Appendix A. Theorem 1 states that the distortion incurred by the proposed procedure is equal to $W_d(P_X, P_{G^\star(Z)})$ up to an additive error term that decays exponentially in $R$, hence converging to $W_d(P_X, P_{G^\star(Z)})$ as $R \to \infty$. Intuitively, as $E$ is no longer rate-constrained asymptotically, we can replace $F$ in (6) by $B \circ E$ and our two-step procedure is equivalent to minimizing (7) w.r.t. $G$, which amounts to minimizing $W_d(P_X, P_{G(Z)})$ w.r.t. $G$ by (6).

Furthermore, according to Theorem 1, the distribution mismatch between $G^\star(B(E(X)))$ and $P_X$ is determined by the quality of the generative model $G^\star$, and is *independent* of $R$. This is natural given that we learn $G^\star$ independently.

We note that the proof of (9) in Theorem 1 hinges upon the fact that $W_d$ is defined w.r.t. the distortion measure $d$. The bound can also be applied to a generator $G'$ obtained by minimizing, e.g., some $f$-divergence [26] between $P_X$ and $P_{G(Z)}$. However, if $W_d(P_X, P_{G'(Z)}) > W_d(P_X, P_{G^\star(Z)})$ (which will generally be the case in practice) then the distortion obtained by using $G'$ will asymptotically be larger than that obtained for $G^\star$. This suggests using $W_d$ rather than $f$-divergences to learn $G$.

## 4  Unsupervised training via Wasserstein++

To learn $G$, $B$, and $E$ from data, we parametrize each component as a DNN and solve the corresponding optimization problems via stochastic gradient descent (SGD). We embed the code $\mathcal{W}$ as vectors (henceforth referred to as "centers") in Euclidean space. Note that the centers can also be learned from the data [6]. Here, we simply fix them to the set of vectors $\{-1, 1\}^R$ and use the differentiable approximation from [9] to backpropagate gradients through this non-differentiable embedding. To ensure that the mapping $B$ is stochastic, we feed noise together with the (embedded) code $E(X)$.

The distribution constraint in (8), i.e., ensuring that $B(E(X)) \sim P_Z$, can be implemented using a maximum mean discrepancy (MMD) [27] or GAN-based [17] regularizer. Firstly, we note that both MMD and GAN-based regularizers can be learned from the samples—for MMD via the corresponding U-estimator, and for GAN via the adversarial framework. Secondly, matching the (simple) prior distribution $P_Z$ is much easier than matching the likely complex distribution $P_X$ as in (3). Intuitively, at high rates, $B$ should learn to ignore the noise at its input and map the code to $P_Z$. On the other hand, as $R \to 0$, the code becomes low-dimensional and $B$ is forced to combine it with the stochasticity of the noise at its input to match $P_Z$. In practice, we observe that MMD is robust and allows to enforce $P_Z$ at all rates $R$, while GAN-based regularizers are prone to mode collapse at low rates.

**Wasserstein++.** As previously discussed, $G^\star$ can be learned via WGAN [16] or WAE [17]. As the WAE framework naturally includes an encoder, it ensures that the structure of the latent space $\mathcal{Z}$ is amenable to encode into. On the other hand, there is no reason that such a structure should emerge

in the latent space of $G$ trained via WGAN (in particular when $\mathcal{Z}$ is high-dimensional).[2] In our experiments we observed that WAE tends to produce somewhat less sharp samples than WGAN. On the other hand, WAE is arguably less prone to mode dropping than WGAN as the WAE objective severely penalizes mode dropping due to the reconstruction error term. To combine the best of both approaches, we propose the following novel combination of the primal and the dual form of $W_d$, via their convex combination

$$W_c(P_X, P_G(Z)) = \gamma \left( \sup_{f \in \mathcal{F}_1} \mathbb{E}_X[f(X)] - \mathbb{E}_Y[f(G(Z))] \right)$$
$$+ (1 - \gamma) \left( \inf_{F:\, F(X) \sim P_Z} \mathbb{E}_X \mathbb{E}_F[d(X, G(F(X)))] \right), \qquad (11)$$

with $\gamma \in [0, 1]$. There are two practical questions remaining. Firstly, minimizing this expression w.r.t. $G$ can be done by alternating between performing gradient updates for the critic $f$ and gradient updates for $G, F$. In other words, we combine the steps of the WGAN algorithm [16, Algorithm 1] and WAE-MMD algorithm [17, Algorithm 2], and call this combined algorithm Wasserstein++. Secondly, one can train the critic $f$ on fake samples from $G(Z)$ or from $G(F(X))$, which will not follow the same distribution in general due to a mismatch between $F(X)$ and $P_Z$, which is more pronounced in the beginning of the optimization process. Preliminary experiments suggest that the following setup yields samples of best quality (in terms of FID score):

(i) Train $f$ on samples from $G(\tilde{Z})$, where $\tilde{Z} = UZ + (1 - U)F(X)$ with $U \sim \text{Uniform}(0, 1)$.

(ii) Train $G$ only on samples from $F(X)$, for both the WGAN and the WAE loss term.

We note that training $f$ on samples from $G(\tilde{Z})$ instead of $G(Z)$ arguably introduces robustness to distribution mismatch in $\mathcal{Z}$-space. A more detailed description of Wasserstein++ can be found in Appendix C, and the relation of Wasserstein++ to existing approaches combining GANs and autoencoders is discussed in Section 6. We proceed to present the empirical evaluation of the proposed approach.

## 5   Empirical evaluation[3]

**Setup.** We empirically evaluate the proposed DPLC framework for $G^\star$ trained via WAE-MMD (with an inverse multiquadratics kernel, see [17]), WGAN with gradient penalty (WGAN-GP) [28], and Wasserstein++ (implementing the 1-Lipschitz constraint in (11) via the gradient penalty from [28]), on two standard generative modeling benchmark image datasets, CelebA [19] and LSUN bedrooms [20], both downscaled to $64 \times 64$ resolution. We focus on these data sets at relatively low resolution as current state-of-the-art generative models can handle them reasonably well, and we do not want to limit ourselves by the difficulties arising with generative models at higher resolutions. The Euclidean distance is used as distortion measure (training objective) $d$ in all experiments.

We measure the quality of the reconstructions of our DPLC systems via mean squared error (MSE) and we assess how well the distribution of the testing reconstructions matches that of the original data using the FID score, which is the recommended measure for image data [18, 29]. To quantify the variability of the reconstructions conditionally on the code $w$ (i.e., conditionally on the encoder input), we estimate the mean conditional pixel variance $\text{PV}[\hat{X}|w] = \frac{1}{N} \sum_{i,j} \mathbb{E}_B[(\hat{X}_{i,j} - \mathbb{E}_B[\hat{X}_{i,j}|w])^2|w]$, where $N$ is the number of pixels of $X$. In other words, PV is a proxy for how well $G \circ B$ picks up the noise at its input at low rates. All performance measures are computed on a testing set of 10k samples held out form the respective training set, except PV which is computed on a subset 256 testing samples, averaged over 100 reconstructions per testing sample (i.e., code $w$).

**Architectures, hyperparameters, and optimizer.** The prior $P_Z$ is an $m$-dimensional multivariate standard normal, and the noise vector providing stochasticity to $B$ has $m$ i.i.d. entries distributed uniformly on $[0, 1]$. We use the DCGAN [30] generator and discriminator architecture for $G$ and $f$, respectively. For $F$ and $E$ we follow [17] and apply the architecture similar to the DCGAN

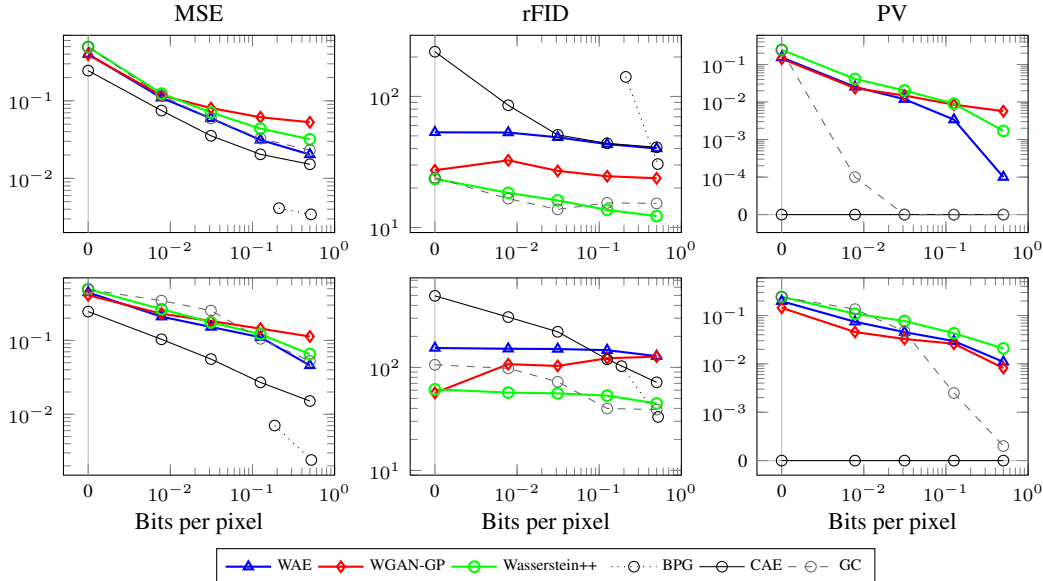

Figure 3: Testing MSE (smaller is better), reconstruction FID (smaller is better), conditional pixel variance (PV, larger is better) obtained by our DPLC model, for different generators $G^\star$, CAE, BPG, as well as GC [15], as function of the bitrate. The results for CelebA are shown in the top row, those for LSUN bedrooms in the bottom row. The PV of our DPLC models steadily increases with decreasing rate, i.e., they generate gradually more image content, as opposed to GC.

discriminator. $B$ is realized as a stack of $n$ residual blocks [31]. We set $m = 128$, $n = 2$ for CelebA, and $m = 512$, $n = 4$ for the LSUN bedrooms data set. We chose $m$ to be larger than the standard latent space dimension for GANs as we observed that lower $m$ may lead to blurry reconstructions.

As baselines, we consider compressive autoencoders (CAEs) with the same architecture $G \circ B \circ E$ but without feeding noise to $B$, training $G, B, E$ jointly to minimize distortion, and BPG [32], a state-of-the-art engineered codec.[4] In addition, to corroborate the claims made on the disadvantages of (3) in Section 3, we train $G \circ B \circ E$ to minimize (3) as done in the generative compression (GC) approach from [15], but replacing $d_f$ by $W_d$.

Throughout, we rely on the Adam optimizer [33]. To train $G$ by means of WAE-MMD and WGAN-GP we use the training parameters form [17] and [28], respectively. For Wasserstein++, we set $\gamma$ in (11) to $2.5 \cdot 10^{-5}$ for CelebA and to $10^{-4}$ for LSUN. Further, we use the same training parameters to solve (8) as for WAE-MMD. Thereby, to compensate for the increase in the reconstruction loss with decreasing rate, we adjust the coefficient of the MMD penalty, $\lambda_{\mathrm{MMD}}$ (see Appendix C), proportionally as a function of the reconstruction loss of the CAE baseline, i.e., $\lambda_{\mathrm{MMD}}(R) = \mathrm{const.} \cdot \mathrm{MSE}_{\mathrm{CAE}}(R)$. We adjust the coefficient $\lambda$ of the divergence term $d_f$ in (3) analogously. This ensures that the regularization strength is roughly the same at all rates. Appendix B provides a detailed description of all architectures and hyperparameters.

**Results.** Table 1 shows sample FID of $G^\star$ for WAE, WGAN-GP, and Wasserstein++, as well as the reconstruction FID and MSE for WAE and Wasserstein++.[5] In Figure 3 we plot the MSE, the reconstruction FID, and PV obtained by our DPLC models as a function of the bitrate, for different $G^\star$, along with the values obtained for the baselines. Figure 1 presents visual examples produced by our DPLC model with $G^\star$ trained using Wasserstein++, along with examples obtained for GC and CAE. More visual examples can be found in Appendix D.

**Discussion.** We first discuss the performance of the trained generators $G^\star$, shown in Table 1. For both CelebA and LSUN bedrooms, the sample FID obtained by Wasserstein++ is considerably smaller than that of WAE, but slightly larger than that of WGAN-GP. Further, Wasserstein++ yields a significantly smaller reconstruction FID than WAE, but a larger reconstruction MSE. Note that the decrease in sample and reconstruction FID achieved by Wasserstein++ compared to WAE should be expected to come at the cost of an increased reconstruction MSE, as the Wasserstein++ objective is obtained by adding a WGAN term to the WAE objective (which minimizes distortion).

We now turn to the DPLC results obtained for CelebA shown in Figure 3, top row. It can be seen that among our DPLC models, the one combined with $G^\star$ from WAE yields the lowest MSE, followed by those based on Wasserstein++, and WGAN-GP. This is not surprising as the optimization of WGAN-GP does not include a distortion term. CAE obtains a lower MSE than all DPLC models which is again intuitive as $G, B, E$ are trained jointly and to minimize distortion exclusively (in particular there is no constraint on the distribution in $\mathcal{Z}$-space). Finally, BPG obtains the overall lowest MSE. Note, however, that BPG relies on several advanced techniques such as entropy coding based on context models (see, e.g., [4, 8–10]), which we did not implement here (but which could be incorporated into our DPLC framework).

Among our DPLC methods, DPLC based on Wasserstein++ attains the lowest reconstruction FID (i.e., its distribution most faithfully reproduces the data distribution) followed by WGAN-GP and WAE. For all three models, the FID decreases as the rate increases, meaning that the models manage *not only to reduce distortion as the rate increases, but also to better reproduce the original distribution*. The FID of CAE increases drastically as the rate falls below 0.03 bpp. Arguably, this can be attributed to significant blur incurred at these low rates (see Figure 9 in Appendix D). BPG yields a very high FID as soon as the rate falls below 0.5 bpp due to compression artifacts.

The PV can be seen to increase steadily for all DPLC models as the rate decreases, as expected. This is also reflected by the visual examples in Figure 1, left: At 0.5 bpp no variability is visible, at 0.125 bpp the facial expression starts to vary, and decreasing the rate further leads to the encoder producing different persons, deviating more and more form the original image, until the system generates random faces.

In contrast, the PV obtained by solving (3) as in GC [15] is essentially 0, except at 0 bpp, where it is comparable to that of our DPLC models. The noise injected into $D = G \circ B$ is hence ignored unless it is the only source of randomness at 0 bpp. We emphasize that this is the case even though we adjust the coefficient $\lambda$ of the $d_f$ term as $\lambda(R) = \text{const.} \cdot \text{MSE}_{\text{CAE}}(R)$ to compensate for the increase in distortion with decreasing rate. The performance of GC in terms of MSE and reconstruction FID is comparable to that of the DPLC model with Wasserstein++ $G^\star$.

We now turn to the DPLC results obtained for LSUN bedrooms. The qualitative behavior of DPLC based on WAE and Wasserstein++ in terms of MSE, reconstruction FID, and PV is essentially the same as observed for CelebA. Wasserstein++ provides the lowest FID by a large margin, for all positive rates. The reconstruction FID for WAE is high at all rates, which is not surprising as the sample FID obtained by WAE is large (cf. Table 1), i.e., WAE struggles to model the distribution of the LSUN bedrooms data set.

For DPLC based on WGAN-GP, in contrast, while the MSE and PV follow the same trend as for CelebA, the reconstruction FID increases notably as the bitrate decreases. By inspecting the corresponding reconstructions (cf. Figure 12 in Appendix D) one can see that the model manages to approximate the data distribution well at zero bitrate, but yields increasingly blurry reconstructions as the bitrate increases. This indicates that either the (trained) function $B \circ E$ is not mapping the original images to $\mathcal{Z}$ space in a way suitable for $G^\star$ to produce crisp reconstructions, or the range of $G^\star$ does not cover the support of $P_X$ well. We tried to address the former issue by increasing the depth of $B$ (to increase model capacity) and by increasing $\lambda_{\text{MMD}}$ (to reduce the mismatch between the distribution of $B(E(X))$ and $P_Z$), but we did not observe improvements in reconstruction quality. We therefore suspect mode coverage issues to cause the blur in the reconstructions.

Finally, GC [15] largely ignores the noise injected into $D$ at high bitrates, while using it to produce stochastic decoders at low bitrates. However, at low rates, the rFID of GC is considerably higher than that of DPLC based on Wasserstein++, meaning that it does not faithfully reproduce the data distribution despite using stochasticity. Indeed, GC suffers from mode collapse at low rates as can be seen in Figure 14 in Appendix D.

Table 1: Reconstruction FID and MSE (without the rate constraint[5]), and sample FID for the trained generators $G^\star$, on CelebA and LSUN bedrooms (smaller is better for all three metrics). Wasserstein++ obtains lower rFID and sFID than WAE, but a (slightly) higher sFID than WGAN-GP.

| | CelebA | | | LSUN bedrooms | | |
|---|---|---|---|---|---|---|
| | MSE | rFID | sFID | MSE | rFID | sFID |
| WAE | 0.0165 | 38.55 | 51.82 | 0.0099 | 42.59 | 153.57 |
| WGAN-GP | / | / | 22.70 | / | / | 45.52 |
| **Wasserstein++** | **0.0277** | **10.93** | **23.36** | **0.0321** | **27.52** | **60.97** |

## 6 Related work

DNN-based methods for compression have become an active area of research over the past few years. Most authors focus on image compression [1–6, 8, 7, 13, 9, 10], while others consider audio [11] and video [12] data. Compressive autoencoders [3, 5, 6, 8, 13, 10] and recurrent neural networks (RNNs) [1, 2, 7] have emerged as the most popular DNN architectures for compression.

GANs have been used in the context of learned image compression before [4, 15, 34, 35]. Work [4] applies a GAN loss to image patches for artifact suppression, whereas [15] applies the GAN loss to the entire image to encourage the decoder to generate image content (but does not demonstrate a properly working stochastic decoder). GANs are leveraged by [36] and [35] to improve image quality of super resolution and engineered compression methods, respectively.

Santurkar et al. [34] use a generator trained with a GAN as a decoder in a compression system. However, they rely on vanilla GAN [14] only rather than considering different $W_d$-based generative models and they do not provide an analytical characterization of their model. Most importantly, they optimize their model using conventional distortion minimization with deterministic decoder, rather than solving the DPLC problem.

Gregor et al. [37] propose a variational autoencoder (VAE)-type generative model that learns a hierarchy of progressively more abstract representations. By storing the high-level part of the representation and generating the low-level one, they manage to partially preserve and partially generate image content. However, their framework is lacking a notion of rate and distortion and does not quantize the representations into a code (apart from using finite precision data types).

Probably most closely related to Wasserstein++ is VAE-GAN [38], combining VAE [24] with vanilla GAN [14]. However, whereas the VAE part and the GAN part minimize different divergences (Kullback-Leibler and Jensen-Shannon in the case of VAE and vanilla GAN, respectively), WAE and WGAN minimize the same cost function, so Wasserstein++ is somewhat more principled conceptually. More generally, learning generative models jointly with an inference mechanism for the latent variables has attracted significant attention, see, e.g., [38–41] and [42] for an overview.

Outside of the domain of machine learning, the problem of distribution-preserving (scalar) quantization was studied. Specifically, [43] studies moment preserving quantization, that is quantization with the design criterion that certain moments of the data distribution shall be preserved. Further, [44] proposes an engineered dither-based quantization method that preserves the distribution of the variable to be quantized.

## 7 Conclusion

In this paper, we studied the DPLC problem, which amounts to optimizing the rate-distortion tradeoff under the constraint that the reconstructed samples follow the distribution of the training data. We proposed different approaches to solve the DPLC problem, in particular Wasserstein++, a novel combination of WAE and WGAN, and analytically characterized the properties of the resulting compression systems. These systems allowed us to obtain essentially artifact-free reconstructions at all rates, covering the full spectrum from learning a generative model of the data at zero bitrate on one hand, to learning a compression system with almost perfect reconstruction at high bitrate on the other hand. Most importantly, our framework improves over previous methods by producing stochastic decoders at low bitrates, thereby effectively solving the DPLC problem for the first time. Future work includes scaling the proposed approach up to full-resolution images and applying it to data types other than images.

**Acknowledgments.** The authors would like to thank Fabian Mentzer for insightful discussions and for providing code to generate BPG images for the empirical evaluation in Section 5.

## Footnotes

[1]Note that a stochastic decoder is necessary if $P_X$ is continuous.

[2]In principle, this is not an issue if $B$ has enough model capacity, but it might lead to differences in practice as the distortion (8) should be easier to minimize if the $\mathcal{Z}$-space is suitably structured, see Section 5.

[3]Code is available at https://github.com/mitscha/dplc.

[4]The implementation from [32] used in this paper cannot compress to rates below $\approx 0.2$ bpp on average for the data sets considered here.

[5]The reconstruction FID and MSE in Table 1 are obtained as $G^\star(F(X))$, *without* rate constraint. We do not report reconstruction FID and MSE for WGAN-GP as its formulation (5) does not naturally include an unconstrained encoder.

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
