[Supplementary Material]

# A  Proof of Theorem 1

We first prove (9). We start by constructing a rate-constrained stochastic function $\hat{F}\colon \mathcal{X} \to \mathbb{R}^m$ as follows. Let $q$ be the nearest neighbor quantizer

$$q(z) = \arg \min_{i \in [2^R]} \|z - c_i\| \tag{12}$$

with the centers $\{c_1, \ldots, c_{2^R}\} \subset R^m$ chosen to minimize $\mathbb{E}[\min_{i \in [2^R]} \|z - c_i\|]$ (the minimum is attained by Prop. 2.1 in [45]). Let $A_i$ be the Voronoi region associated with $c_i$, i.e., $A_i = \{z \in \mathbb{R}^m \colon \|z - c_i\| = \min_{i \in [2^R]} \|z - c_i\|\}$. We now set $E(X) = q(F^\star(X))$ and $B(i) = Z_i$, where $F^\star$ is a minimizer of (7) for $G = G^\star$ and $Z_i \sim P_{Z|Z \in A_i}$, independent of $Z$ given $A_i$. It holds $\hat{F}(X) := B(E(X)) \sim Z$ by construction and the described choice of $B, E$ is feasible for (8). We continue by upper-bounding $d(X, G^\star(\hat{F}(X)))$

$$
\begin{aligned}
d(X, G^\star(\hat{F}(X))) &\leq d(X, G^\star(F^\star(X))) + d(G^\star(F^\star(X)), G^\star(\hat{F}(X))) \\
&\leq d(X, G^\star(F^\star(X))) + K\|F^\star(X) - \hat{F}(X)\| \\
&\leq d(X, G^\star(F^\star(X))) + K\|F^\star(X) - c_{E(X)}\| + K\|c_{E(X)} - \hat{F}(X)\|.
\end{aligned} \tag{13}
$$

By Cor. 6.7 from [46] we have

$$\mathbb{E}_X[\|F^\star(X) - c_{E(X)}\|] \leq 2^{-\frac{R}{m}}(C_1 \mathbb{E}[\|Z\|^{1+\delta}] + C_2), \; \delta > 0, \; 2^R > C_3, \tag{14}$$

where $C_1, C_2, C_3 > 0$ are numerical constants depending on $\delta, m$ and $\|\cdot\|$. The same upper bound holds for $\mathbb{E}[\|c_{E(X)} - \hat{F}(X)\|]$ as $E(X) = q(F^\star(X)) = q(\hat{F}(X))$, i.e., $\|c_{E(X)} - \hat{F}(X)\| = \|c_{q(\hat{F}(X))} - \hat{F}(X)\|$, and $\hat{F}(X) \sim Z \sim F^\star(X)$. Taking the expectation on both sides of (13) and using (14) in the resulting expression yields the upper bound in (9). The lower bound is obtained by noting that the minimization (7) includes the rate-constrained mappings $B \circ E$.

Eq. (10) directly follows from the distribution constraint in (8), $B(E(X)) \sim P_Z$: This constraint implies $G^\star(B(E(X))) \sim G^\star(Z)$ which in turn implies (10).

**Remark 1.** *The construction of the discrete encoder $\hat{F}$ in the proof of Theorem 1 requires optimal vector quantization, which is generally NP hard. However, if we make stronger assumptions on $P_Z$ than done in the statement of Theorem 1 one can prove exponential convergence without optimal vector quantization. For example, if $Z$ is uniformly distributed on $[0,1]^m$, $R = km$ for a positive integer $k$, and $\|\cdot\|$ is the Euclidean norm, then one can partition $[0,1]^m$ into $2^R$ hypercubes of equal edge length $1/2^k = 1/2^{\frac{R}{m}}$ and associate the centers $c_i$ with the centers of these hypercubes. In this case, the two last terms in (13) are upper-bounded by $\sqrt{m} \cdot 2^{-\frac{R}{m}}$. The quantization error thus converges to 0 as $2^{-\frac{R}{m}}$ for $R \to \infty$.*

# B  Hyperparameters and architectures

**Learning the generative model $G^\star$.** The training parameters used train $G^\star$ using WAE, WGAN-GP, and Wasserstein++ are shown in Table 2. The parameters for WAE correspond to those used for the WAE-MMD experiments on CelebA in [17], see Appendix C.2, with the difference that we use a batch size of 256 and a slightly modified schedule (note that 41k iterations with a batch size of 256 correspond to roughly 55 epochs with batch size 100, which is suggested by [17]). This does not notably impact the performance WAE (we obtain a slightly lower sample FID than reported in [17, Table 1]). The parameters for WGAN-GP correspond to those recommended for LSUN bedrooms in [28, Appendix E].

**Learning the function $B \circ E$.** The training parameters to solve (8) can be found in Table 3. To solve (1) (i.e., to train the CAE baseline) we use the same parameters as for WAE (except that $\lambda_{\text{MMD}} = 0$ as there is no distribution constraint in (1)), see Table 2.

**Training the baseline (3) as in [15].** To solve (3) we use the parameters and schedule specified in Table 2 for Wasserstein++ (except that we do not need $\lambda_{\text{MMD}}$), and we determine $\lambda$ in (3) based on the bitrate as $\lambda(R) = 2.5 \cdot 10^{-5} \cdot \frac{\text{MSE}_{\text{CAE}}(R)}{\text{MSE}_{\text{CAE}}(0.5\text{bpp})}$ for CelebA and $\lambda(R) = 7.5 \cdot 10^{-5} \cdot \frac{\text{MSE}_{\text{CAE}}(R)}{\text{MSE}_{\text{CAE}}(1\text{bpp})}$ for LSUN bedrooms.

**Architectures.** We use the following notation. `cxsy-z` stands for a 2D convolution with an $x \times x$ kernel, stride `y`, and `z` filters, followed by the ReLU non-linearity (the ReLU non-linearity is omitted when the convolution is followed by quantization or the tanh non-linearity). The suffixes `b` and `l`, i.e., `cxsyb-z` and `cxsyl-z`, indicate that batch normalization is employed before the non-linearity and layer normalization as well as leaky ReLU with a negative slope of $0.2$ instead of ReLU, respectively. `txsyb-z` stands for a transposed 2D convolution with an $x \times x$ kernel, stride $1/y$, and `z` filters, followed by batch normalization and ReLU non-linearity. `fc-z` denotes flattening followed by a fully-connected layer with `z` neurons. `r-z` designates a residual block with `z` filters in each layer. The abbreviations `bn`, `tanh`, and `-q` are used for batch normalization, the tanh non-linearity, and quantization with differentiable approximation for gradient backpropagation, respectively. $k$ is the number of channels of the quantized feature representation (i.e., $k$ determines the bitrate), and the suffix `+n` denotes concatenation of an $m$-dimensional noise vector with i.i.d. entries uniformly distributed on $[0, 1]$, reshaped as to match the spatial dimension of the feature maps in the corresponding network layer.

- $F$: `c4s2-64, c4s2b-128, c4s2b-256, c4s2b-512, fc-`$m$`, bn`

- $G$: `t4s2b-512, t4s2b-256, t4s2b-128, t4s2b-64, t4s2b-64, c3s1-3, tanh`

- $f$: `c3s1-64, c4s2l-64, c4s2l-128, c4s2l-256, c4s2l-512, fc-1`

- $E$: `c4s2-64, c4s2b-128, c4s2b-256, c4s2b-512, c3s1-`$k$`-q+n`

- $B$: `c3s1-512, r-512, ..., r-512, fc-`$m$`, bn`

Table 2: Adam learning rates $\alpha_F, \alpha_G, \alpha_f$ for the WAE encoder $F$, the generator $G$, and the WGAN critic $f$, respectively, Adam parameters $\beta_1, \beta_2$, MMD regularization coefficient $\lambda_{\text{MMD}}$, mini-batch size $b$, number of (generator) iterations $n_{\text{iter}}$, and learning rate schedule (LR sched.), for CelebA. The number of critic iterations per generator iteration is set to $n_{\text{critic}} = 5$ for WGAN-GP and Wasserstein++. **For LSUN bedrooms**, the parameters are identical, except that $\lambda_{\text{MMD}} = 300$ for WAE and Wasserstein++, and the number of iterations is doubled (with the learning rate schedule scaled accordingly) for all three algorithms.

| | $\alpha_F$ | $\alpha_G$ | $\alpha_f$ | $\beta_1$ | $\beta_2$ | $\lambda_{\text{MMD}}$ | $\lambda_{\text{GP}}$ | $b$ | $n_{\text{iter}}$ | LR sched. |
|---|---|---|---|---|---|---|---|---|---|---|
| WAE | $10^{-3}$ | $10^{-3}$ | / | 0.5 | 0.999 | 100 | / | 256 | 41k | ×0.4@22k;38k |
| WGAN-GP | / | $10^{-4}$ | $10^{-4}$ | 0.5 | 0.900 | | / | 10 | 64 | 100k | / |
| Wasserstein++ | $3 \cdot 10^{-4}$ | $3 \cdot 10^{-4}$ | $10^{-4}$ | 0.5 | 0.999 | 100 | 10 | 256 | 25k | ×0.4@15k;21k |

Table 3: Adam parameters $\alpha, \beta_1, \beta_2$, MMD regularization coefficient $\lambda_{\text{MMD}}$ as a function of the MSE incurred by CAE at rate $R$ ($\text{MSE}_{\text{CAE}}(R)$), mini-batch size $b$, number of iterations $n_{\text{iter}}$, and learning rate schedule (LR sched.) used to solve (8).

| | $\alpha$ | $\beta_1$ | $\beta_2$ | $\lambda_{\text{MMD}}(R)$ | $b$ | $n_{\text{iter}}$ | LR sched. |
|---|---|---|---|---|---|---|---|
| CelebA | $10^{-3}$ | 0.5 | 0.999 | $150 \cdot \frac{\text{MSE}_{\text{CAE}}(R)}{\text{MSE}_{\text{CAE}}(0.5\text{bpp})}$ | 256 | 41k | ×0.4@22k;38k |
| LSUN bedrooms | $10^{-3}$ | 0.5 | 0.999 | $800 \cdot \frac{\text{MSE}_{\text{CAE}}(R)}{\text{MSE}_{\text{CAE}}(1\text{bpp})}$ | 256 | 82k | ×0.4@44k;76k |

# C   The Wasserstein++ algorithm

---

**Algorithm 1** Wasserstein++

---

**Require:** MMD regularization coefficient $\lambda_{\text{MMD}}$, WGAN coefficient $\gamma$, WGAN gradient penalty coefficient $\lambda_{\text{GP}}$, number of critic iterations per generator iteration $n_{\text{critic}}$, mini-batch size $b$, characteristic positive-definite kernel $k$, Adam parameters (not shown explicitly).

1: **Initialize** the parameters $\phi$, $\theta$, and $\psi$ of the WAE encoder $F_\phi$, the generator $G_\theta$, and the WGAN discriminator $f_\psi$, respectively.

2: **while** $(\phi, \theta, \psi)$ not converged **do**

3:     **for** $t = 1, \ldots, n_{\text{critic}}$ **do**

4:         Sample $\{x_1, \ldots, x_b\}$ from the training set

5:         Sample $\bar{z}_i$ from $F_\phi(x_i)$, for $i = 1, \ldots, b$

6:         Sample $\{z_1, \ldots, z_b\}$ from the prior $P_Z$

7:         Sample $\{\eta_1, \ldots, \eta_b\}$ from $\text{Uniform}(0, 1)$

8:         Sample $\{\nu_1, \ldots, \nu_b\}$ from $\text{Uniform}(0, 1)$

9:         $\tilde{z}_i \leftarrow \eta_i z_i + (1 - \eta_i)\bar{z}_i, \quad \text{for } i = 1, \ldots, b$

10:        $\hat{x}_i \leftarrow G_\theta(\bar{z}_i), \quad \text{for } i = 1, \ldots, b$

11:        $\tilde{x}_i \leftarrow \nu_i x_i + (1 - \nu_i)\hat{x}_i, \quad \text{for } i = 1, \ldots, b$

12:        $L_f \leftarrow \frac{1}{b}\sum_{i=1}^{b} f_\psi(\hat{x}_i) - f_\psi(x_i) + \lambda_{\text{GP}}(\|\nabla_{\tilde{x}_i} f_\psi(\tilde{x}_i)\| - 1)^2$

13:        $\psi \leftarrow \text{Adam}(\psi, L_f)$

14:     **end for**

15:     $L_d \leftarrow \frac{1}{b}\sum_{i=1}^{b}\|x_i - G_\theta(\bar{z}_i)\|$

16:     $L_{\text{MMD}} \leftarrow \frac{1}{b(b-1)}\sum_{\ell \neq j} k(z_\ell, z_j) + \frac{1}{b(b-1)}\sum_{\ell \neq j} k(\bar{z}_\ell, \bar{z}_j) - \frac{2}{b^2}\sum_{\ell,j} k(z_\ell, \bar{z}_j)$

17:     $L_{\text{WGAN}} \leftarrow \frac{1}{b}\sum_{i=1}^{b} -f_\psi(G_\theta(\bar{z}_i))$

18:     $\theta \leftarrow \text{Adam}(\theta, (1 - \gamma)L_d + \gamma L_{\text{WGAN}})$

19:     $\phi \leftarrow \text{Adam}(\phi, (1 - \gamma)(L_d + \lambda_{\text{MMD}}L_{\text{MMD}}))$

20: **end while**

---

# D   Visual examples

In the following, we show random samples and reconstructions produced by different DPLC models and CAE, at different bitrates, for the CelebA and LSUN bedrooms data set. None of the examples are cherry-picked.

WAE

WGAN-GP

Wasserstein++

Figure 4: Random samples produced by the trained generator $G^\star(Z)$, with $Z \sim P_Z$, on CelebA. The samples produced by WGAN-GP and Wasserstein++ are sharper than those generated by WAE.

Figure 5: Testing reconstructions produced by our DPLC model with WAE $G^\star$, along with the original image (green border), for CelebA. The variability between different reconstructions increases as the bitrate decreases.

Figure 6: Testing reconstructions produced by our DPLC model with WGAN-GP $G^\star$, along with the original image (green border), for CelebA. The variability between different reconstructions increases as the bitrate decreases.

Figure 7: Testing reconstructions produced by our DPLC model with Wasserstein++ $G^\star$, along with the original image (green border), for CelebA. The variability between different reconstructions increases as the bitrate decreases.

Figure 8: Testing reconstructions produced using $G \circ B \circ E$ obtained by solving (3) similarly as in GC [15], along with the original image (green border), for CelebA. There is no variability between different reconstructions except at 0 bpp.

0.000 bpp          0.008 bpp          0.031 bpp

0.125 bpp          0.500 bpp          original

Figure 9: Testing reconstructions produced by CAE, for CelebA. The reconstructions become increasingly blurry as the rate decreases.

WAE

WGAN-GP

Wasserstein++

Figure 10: Random samples produced by the trained generator $G^\star(Z)$, with $Z \sim P_Z$, for LSUN bedrooms. The samples produced by WGAN-GP and Wasserstein++ are sharper than those generated by WAE.

Figure 11: Testing reconstructions produced by our DPLC model with WAE $G^{\star}$, along with the original image (green border), for LSUN bedrooms. The reconstructions are blurry at all rates.

Figure 12: Testing reconstructions produced by our DPLC model with WGAN-GP $G^\star$, along with the original image (green border), for LSUN bedrooms. The reconstructions are blurry at all rates except at 0 bpp.

Figure 13: Testing reconstructions produced by our DPLC model with Wasserstein++ $G^\star$, along with the original image (green border), for LSUN bedrooms. The variability between different reconstructions increases as the bitrate decreases. The reconstructions are quite sharp at all rates.

Figure 14: Testing reconstructions produced using $G \circ B \circ E$ obtained by solving (3) similarly as in GC [15], along with the original image (green border), for LSUN bedrooms. The method produces a stochastic decoder at very low rates but suffers from mode collapse.

Figure 15: Testing reconstructions produced by CAE, for LSUN bedrooms. The reconstructions become increasingly blurry as the rate decreases.