[Reviews · NeurIPS 2018]

Reviewer 1



The paper proposes a novel problem formulation for lossy compression, namely distribution-preserving lossy compression (DPLC). For a rate constrained lossy compression scheme, for large enough rate of the compression scheme, it is possible to (almost) exactly reconstruct the original signal from its compressed version. However, as the rate gets smaller, the reconstructed signal necessarily has very high distortion. The DPLC formulation aims to alleviate this issue by enforcing an additional constraint during the design of the encoder and the decoder of the compression scheme. The constraint requires that irrespective of the rate of the compression the distribution of the reconstructed signal is the same as that of the original signal. This ensures that for the very small rate the decoder simply outputs a new sample from the source distribution and not a useless signal with very large distortion. In particular, a DPLC scheme needs to have a stochastic decoder which is hard to train using the existing techniques in the literature. The authors propose a way to deal with this and employ Wasserstein distance to measure the divergence between the original signal distribution and that of the output of the decoder. The authors build upon the recent line of work on utilizing neural networks to model the signal distributions. The authors rely on both generative adversarial networks and autoencoders to design a DPLC scheme, which gives a novel object called Wassserstein++. The authors theoretical quantify the performance of their proposed procedure for encoder and decoder design by relating its distortion performance to a suitable Wasserstein distance. They further, implement their procedure with two GAN datasets and show that it indeed allows them to map between the exact reconstruction and sampling from the generative model as the rate of the compression scheme approaches zero. Overall, I find the problem formulation unique with a reasonably comprehensive treatment of the problem presented in the paper. The paper is well written. Minor comment: It's not clear to me what $G$ refers to in the statement of Theorem 1. Should it be $G^{\star}$? Line 148: ".....$F$ in (6)....." should be "......$F$ in (7)....."?

Reviewer 2



Summary ======= This paper proposes a formulation of lossy compression in which the goal is to minimize distortion subject to the constraint that the reconstructed data follows the data distribution (and subject to a fixed rate). As in Agustsson et al. (2018), a stochastic decoder is used to achieve this goal, but trained for a combination of WGAN/WAE instead of LS-GAN. The encoder is trained separately by minimizing MMD. The authors furthermore provide a theorem which provides bounds on the achievable distortion for a given bit rate. Good ==== The proposed formulation is a slightly different take on lossy compression, which has the potential to inspire novel solutions. Constraining the reconstruction distribution – although difficult in practice – would ensure that reconstructions are always of high perceptual quality. Bad === The empirical evaluation is quite limited. In particular, the comparison is limited to toy datasets (CelebA, LSUN). It would have been nice to see how the proposed approach performs compared to other published approaches or standardized codecs on more realistic datasets (e.g., Kodak). Furthermore, why were qualitative comparisons with Agustsson et al. (2018) omitted in Figure 1 when they were clearly available (Figure 2)? Despite being a more complicated approach than Agustsson et al. (2018), it does not seem to perform better (Figure 2). The only demonstrated difference is an increase in conditional variance, which in itself may not be a desirable property for a practical compression system nor does it provide strong evidence that the reconstruction follows the target distribution more closely. The paper seems to lack important details. For example, how is the rate limitation of the encoder achieved? How is the output of the encoder quantized? In Figure 2, how are the quantized outputs encoded into bits? I find it quite difficult to follow the train of thought on page 4. Please provide more detail on why joint optimization of G and F is worse than optimizing G first and subsequently optimizing F (L128-L137), in particular elaborate the claim that "constraining F implies that one only optimizes over a subset of couplings." This constraint still seems in place in the two-step optimization (Equation 8), so why is it not an issue then? The role and implications of Theorem 1 could be explained more clearly. The authors claim that "Theorem 1 states that the distortion incurred by the proposed procedure is equal to W(P_X, P_G*) [...]". However, this assumes that we solve Equation 8 exactly, which may be as difficult as solving the compression problem itself, so the bound isn't very practical. It seems to me the main purpose is therefore to show that the achievable distortion is limited by the Wasserstein distance of G*, which justifies minimizing the Wasserstein distance of G* via WGAN/WAE. Correct me if I missed something. Minor ===== Constraining the perceptual quality to be perfect limits the achievable distortion (Blau & Michaeli, 2017). In some applications it may be desirable to trade off perceptual quality for lower distortion, which is not possible with the proposed formulation.

Reviewer 3



This works uses generative models for the task of compression. The premise behind this paper is that distribution preserving lossy compression should be possible at all bitrates. At low bitrates, the scheme should be able to recover data from the underlying distribution, while at higher bitrates it produces data (images) that reconstructs the input samples. That the scheme works as advertised can be seen rather convincingly in section 5 containing the evaluations. The paper is well written and constitutes a very interesting exercise. The compression setup consists of an encoder to compress the input image E, and a stochastic component B, and a generator G recovers the input. Together, they are trained to minimize a distortion measure. The objective function is formulated as a Lagrangian consisting of the distortion measure between input and reconstruction and a divergence term, implemented as a GAN which takes effect at low bitrates so that it recovers images from the underlying distribution. Several other generative compressive setups are compared against. But the closest comparison (in terms of approach) seems to be the work "Generative Adversarial Networks for learned image compression" (termed GC in the results). In this connection, they point out that if one were simply to minimize the distortion measure as done in GC, it results in the setup largely ignoring the noise injected except at very low bitrates where it is the only factor involved. I think I can intuitively understand this part, but a little clarification would be useful. The reasoning in section 3, on why one cannot simply replace the distortion metric using a Wasserstein distance seems to be that the encoder is rate constrained and therefore optimizing this metric would not result in optimizing over all couplings. They then go on to decompose the distortion measure (equation (4)) as the sum of two terms, one of which is the wasserstein distance and the other, a term containing the distortion rate which goes to zero at large values.. By doing so (the generator is learnt separately) they can replace the distortion measure with the said Wasserstein term and optimize it. The connection where they jointly train the setup to function essentially as a VAE-GAN by combining the two terms in the objective - the distortion as the primal, and the divergence as the dual is a most interesting trick for future use. One aspect that I could not piece together (it is likely that I am missing it) is how the Wasserstein++ algorithm manages to enforce the 1-Lipshitz constraint for the GAN part of the objective. The 1-Lipshitz requirement (as one would expect) is explicitly mentioned in equation 11 for the GAN term. As far as I can see, weight clipping/gradient penalty does not seem to be mentioned in Wasserstein++. As regards the evaluation studies produced in section 5, I am curious about use of the Frechet Inception Distance as a metric for use in GANs. On the whole, I am very impressed with this work. It presents a way to attain generative compression at both low and high bitrates, with reconstructions getting better at higher bitrates and the model recovering distribution images (in GAN like fashion) at low bitrates. The use of Wasserstein autoencoders is also quite interesting. Answers to specific review criteria below: 1) Quality: Excellent. 2) Clarity: Very good. The mathematical details could perhaps be worked out as an extended tutorial. 3) Originality: Very good. The material presents Wasserstein autoencoders and GANs (combined as Wasserstein++) as the approach to effect compression. This is rather unique. 4) Significance: Very good. Generative compression is a significant area of research.